# The kinematics of cyclic human movement

**Manfred M. Vieten** *, **Christian Weich**

Department of Sport Science, University of Konstanz, Konstanz, Germany

* Manfred.Vieten@uni-konstanz.de

## Abstract

Literature mentions two types of models describing cyclic movement—theory and data driven. Theory driven models include anatomical and physiological aspects. They are principally suitable for answering questions about the reasons for movement characteristics, but they are complicated and substantial simplifications do not allow generally valid results. Data driven models allow answering specific questions, but lack the understanding of the general movement characteristic. With this paper we try a compromise without having to rely on anatomy, neurology and muscle function. We hypothesize a general kinematic description of cyclic human motion is possible without having to specify the movement generating processes, and still get the kinematics right. The model proposed consists of a superposition of six contributions–subject's attractor, morphing, short time fluctuation, transient effect, control mechanism and sensor noise, while characterizing numbers and random contributions. We test the model with data from treadmill running and stationary biking. Applying the model in a simulation results in good agreement between measured data and simulation values. We find in all our cases the similarity analysis between measurement and simulation is best for the same subjects—$\delta_{run}^{same\ sub} > 55\%$ and $\delta_{bike}^{same\ sub} > 64\%$. All comparisons between different subjects are $51\% > \delta_{run}^{different\ sub}$ and $52\% > \delta_{bike}^{different\ sub}$. This uniquely allows for the identification of each measurement for the associated simulation. However, even different subject comparisons show good agreement between measurement and simulation results of differences $\delta_{run} = 6.7 \pm 4.7\%$ and $\delta_{bike} = 5.1 \pm 4.5\%$.

## Introduction

Bipedal gait, especially walking, has been the most decisive development of homo sapiens to surpass their ancestors and relatives [1]. In the past centuries further cyclic motions like swimming, cycling, rowing or skiing came along, to overcome natural obstacles, to facilitate traveling and then as leisure activities. Recently, cyclic motion descriptions have served as biological templates for developments in robotics together with developments in artificial intelligence [2]. Although cyclic movements are performed a thousand-fold each day in everyday life, their underlying composition and structure is not fully understood.

The kinematics of human cyclic motion seems rather simple at first glance. Detailed observations display a repeating structure and some fluctuation producing similar but not identical repetitive cycles of movements [3, 4]. These changes often describe a transient effect at the

**Data Availability Statement:** All data files are available from zenodo.org under the direct link http://doi.org/10.5281/zenodo.3518415.

**Funding:** AFF-grand "cyclic human motion - 2019" of the University of Konstanz. The funders had no role in study design, data collection and analysis,

decision to publish, or preparation of the manuscript.

**Competing interests:** The authors have declared that no competing interests exist.

start of the movement [4–6], as generally observed in dynamical systems [7, 8]. Moreover, various perturbations alter the regularity of the ongoing movement and stride time dynamics [9–12]. Dingwell and Kang [13] describe these findings as ′inherent biological noise′, being local instabilities [14] during movements like walking, without causing falls or stumbles, meaning that the subjects move ′orbitally stable′. Nashner [15] pointed out, that the described continuity after perturbations is retained by adjusting parameters of the present walking motion rather than recruiting a new motor pattern (p. 650).

Modern quantitative scientific endeavors to understand the mechanism behind the central movement trait already began as early as in the nineteenth century [16]. Describing cyclic motion most often is realized by selected specific body markers and their coordinate portrayal as function of time [17]. The classical gait parameters such as step length, step frequency, velocity as well as marker tracking from digitizing systems carry most of the information considered. With the advent of direct acceleration measurement further and subtler information, which coordinate explanation cannot deliver, can serve to describe cyclic motion. Coordinate data, however, can at least in principle, be generated from acceleration data by two consecutive integrations with respect to time. However, integration is a smoothing process, which makes it evident, that important information gets lost.

For this reason, we propose a *mathematical model of the kinematic of the human cyclic motion* based on acceleration data. It allows simulation of cyclic movement and comparison with measured data. We illustrate this model as a superposition of six mathematical terms covering the motion as a (1) limit-cycle attractor, (2) individual attractor morphing, (3) short time random fluctuation in form of "random walk", (4) the transient effect describing initial oscillations around the attractor at the onset of the activity subsiding with increasing time, (5) a control process being activated when stride variations tend to exceed the morphed attractors' boundaries, and (6) the influence of noise generated by the measurement device—accelerometers. Thus, this model allows extension of earlier findings specifically about the variability of subjects' cyclic movement with its fixed and random components.

There exist two types of models describing human cyclic motion—theory driven and data driven [18]–both with its own strong and weak aspects. For example, a theory driven model as described by Gerritsen et al. [19] gives insight into the working of seven muscle groups within the lower extremities. The necessity of keeping the model manageable, in the mentioned paper by using a 2-dimensional rigid body model, leads to deviations from the actual movement. On the other hand the data driven model of Janssen et al. [18] was able to detect the influence of emotions onto the movement pattern. They applied deep machine learning by using artificial neural nets, allowing identification of subtle effects. While here the detection movement characteristics caused by emotions is nicely achieved, the specifics of the gait changes remained undetected. With the present paper we attempt a compromise, by not having to rely on anatomy and muscle function, but still trying to understand kinematic processes and the movement pattern quantitatively. A study on cycling at two different power outputs (150 W and 300 W) at a cadence of 90 rpm [20] found differences in the muscle activities detected via EMG, while kinematic data stayed almost unchanged. This result together with the stability of the individual's attractor over time and after rehabilitation [21, 22] is motivation to examine the possibility to quantitatively describe movement without the knowledge of muscle activity.

The purpose of this paper is to precisely outline the kinematics of cyclic motion by establishing the necessary mathematical equations, which allow simulation. The method presented permits identifying subject specific movement constants. The testing of model and method is done on two classical cyclic motions: running (on a treadmill) and (stationary) biking.

## Method

The first section "Model" of this paragraph is devoted to the details of our model. The six contributing terms are specified with their deterministic and probabilistic components. Following in the section "Model's characteristic constants" we show how $\delta M$, the mean distance between two attractors, is calculated and how this parameter allows determination of the model's characteristic constants. To see how measurements are fitted into to model the section "Data handling" makes the connection between the raw acceleration data and the specific input format to the model. One of our objectives is to quantify the similarity/dissimilarity of an attractor compared to another attractor, which is not influenced by the transient effect and by morphing, changing one attractor into another. Such an attractor we call a super attractor. Its construct is given in the section "Super attractor" and used in the section "Similarity analysis" to quantify how similar the super attractor is compared to a tested one. In the section "Separating the transient effect form morphing" the super attractor is used again to achieve the separation. In the section "Simulation" some settings are specified and a link on the internet to the used computer apps is given. Finally, the necessary information on the "Subjects", the "Equipment", the "Running data" and the "Cycling data" is presented.

### Model

We construct the full acceleration $\overrightarrow{K}(t)$ as a superposition of the six terms

$$\overrightarrow{K}(t) = \overrightarrow{A}(t) + \overrightarrow{M}(t) + \overrightarrow{F}(t) + \overrightarrow{T}(t) + \overrightarrow{C}(t) + \overrightarrow{N}(t) \tag{1}$$

1. $\overrightarrow{A}(t)$ the Limit-Cycle-**A**ttractor, a constant acceleration pattern being repeated with every cycle.

2. $\overrightarrow{M}(t)$ attractor **M**orphing, which allows minor deviations from the actual attractor values.

3. $\overrightarrow{F}(t)$ short time **F**luctuation in form of a "random walk".

4. $\overrightarrow{T}(t)$ **T**ransient effect, which can be present at the start and decreases rapidly.

5. $\overrightarrow{C}(t)$ **C**ontrol mechanism, kicking in when actual accelerations deviate too much from the morphed attractor.

6. $\overrightarrow{N}(t)$ **N**oise caused by the accelerometers.

1. Limit-Cycle-Attractor $\overrightarrow{A}(t)$ can be regarded as the average of all cycles. This however, is an idealized definition, which cannot fully be met, since this would call for averaging of an infinite number of cycles. Instead, we approximate the attractor by a finite number of cycles, which for later examples we chose the number of complete cycles within a specified minute of the data collection.

$$\overrightarrow{A}\left[\tau_j\right](t) = \frac{1}{n}\sum_{i=1}^{n}\overrightarrow{a}\left(i \cdot \tau_j\right) \tag{2}$$

is a closed line in 3D acceleration space with $\overrightarrow{a}$ the measured acceleration and $j$ being the number of consecutive data points within an attractor. Such an approximated attractor is

characteristic for each individual [21, 22]. The actual calculation starts with dividing each data set into one-minute sections and calculation of the attractors [23]. There is one important methodological difference however. Instead of adding the cycles, which have different numbers of data points in temporal order, we describe each cycle as consisting of a fixed data point number n. This is achieved through spline approximation. The number n stands for the mean number of data points of all complete cycles within a one-minute interval. So, we treat each cycle as lasting an identical time interval equal to the mean cycle duration. Afterwards we add up all cycle values for each of the n points and divide them by the number of cycles. The results represent mean values of the one-minute data sets preserving the original sampling frequency, while still containing the influence of morphing, random walk, transient effect and the control mechanism. The data set least influenced serves as attractor to compare all others with. Appropriate attractors are those for which time $t \gg t_T$ (transient time, explained below).

2. A time-dependent individual attractor morphing $\overrightarrow{M}(t)$ is described as the attractor change from start $t_S$ to end $t_E$ minute by minute. The equation is of heuristic nature. It must be capable of describing the changes of a given attractor and its development to the final attractor as a function of time. We take care of this process by taking attractor approximations at beginning $\overrightarrow{A}(t_S)$ and end $\overrightarrow{A}(t_E)$ and describe the morphing of the two attractor approximations, introducing the three dimensionless constants $a_0$, $a_1$, $a_2$, by

$$\overrightarrow{M}(t) = (\overrightarrow{A}(t_S) - \overrightarrow{A}(t_E)) \cdot a_0 \cdot \left\{ \sqrt{\frac{(t_E - t)}{t_E}} + a_1 \cdot sin\left( a_2 \cdot 2\pi \frac{(t_E - t)}{t_E} \right) \right\} \qquad (3)$$

Important to mention: The morphing is small compared to attractor differences between individuals.

3. Fluctuation $\overrightarrow{F}(t)$ in the form of a "random walk". These are changes around a morphed attractor described with the iteration

$$\overrightarrow{F}(t) = \overrightarrow{F}\left(\frac{l}{f_S}\right) = \overrightarrow{F}\left(\frac{l-1}{f_S}\right) + RN[0, \sigma_{RW}] \cdot \begin{pmatrix} sin(\vartheta(l)) \cdot cos(\varphi(l)) \\ sin(\vartheta(l)) \cdot sin(\varphi(l)) \\ cos(\vartheta(l)) \end{pmatrix} \qquad (4)$$

Here, $l$ is the data number. An aberration from the attractor can happen in any direction. We describe this using the angles $\vartheta$ and $\varphi$. Their actual values are random having a uniform distribution on the sphere with the polar and azimuthal angles:

$$\begin{aligned} \vartheta(l) &= RU[0, \pi] \\ \varphi(l) &= RU[0, 2\pi] \end{aligned} \qquad (5)$$

$RU[\alpha,\beta]$ represents random generation with a uniform characteristic within the interval [$\alpha$, $\beta$]. With this definition the standard deviation of the random walk depends on the sampling frequency $f_S$. Since the random walk must not be dependent on the specifics of a measurement–the sampling frequency $f_S$ -, we introduce a parameter $\phi$ (random walk's strength), which does not change with the sampling frequency.

$$\sigma_{RW} = \frac{f_S}{10^6} \cdot \phi \qquad (6)$$

The factor $10^6$ was introduced for convenience. For simulating the movement $\phi$ together with $\overrightarrow{C}(t)$ (see below) must be chosen to reproduce the statistical spread of the data around the attractor.

4. The **C**ontrolling mechanism $\vec{C}(t)$, respectively the vector component $C_k(t)$, is kicking in when the distance to the morphed attractor's coordinates passes the border $b_k$

$$b_k(j) = b \cdot \frac{\sigma_k(j)}{\langle \sigma_k \rangle} \tag{7}$$

at attractor point $j$. Here $b$ is the controlling constant and $\sigma_k(j)$ the attractor's standard deviation, which is divided by the average of the attractor's deviation $\langle \sigma_k \rangle$. This takes care of the changing width of the acceleration bundle. The correction term, being activated at time $t_b$, is modeled as

$$C_k(t, t_b) = -\int\limits_{t_b}^{t} RN[1, \sigma_M](t') \cdot A_k \cdot \frac{(t' - t_b)}{\tau} \cdot e^{-\frac{(t'-t_b)}{\tau}} \cdot sign(F_k(t_b)) \cdot \Theta(t_M + t_b - t') \cdot dt' \tag{8}$$

With $sign(\ldots)$ being the signum and $\Theta(\ldots)$ the step function. We set the maximal acceleration change to $\tau = 80$ $ms$ analogous to the style of a muscle's timely response [24] with acceleration effectively lasting $t_M = 4 \cdot \tau = 320$ $ms$, to obtain

$$b_k = \int\limits_{t_b}^{t_M + t_b} A_k \cdot \frac{t - t_b}{\tau} \cdot e^{-\frac{t - t_b}{\tau}} \, dt \tag{9}$$

$b_k$ is the acceleration necessary to get back precisely onto the morphed attractor values. This holds true for

$$A_k = \frac{b_k}{\tau - (\tau + t_M) \cdot e^{\frac{-t_M}{\tau}}} \tag{10}$$

$RN[1, \sigma_M](t)$ represents a normally distributed random element introducing some deviation from a perfect working controlling mechanism.

5. The transient effect $\vec{T}(t)$ is a temporary oscillation around the attractor at the beginning of a cyclic movement. The starting value of the oscillation might be very individual, specific to the subject, and having a part of the starting value occurring by sheer chance. We model the deviation as the solution of a damped harmonic oscillator, where the transient term can be viewed as the departure from the morphed attractor

$$\vec{T}(t) = \left[ \sum_{h=1}^{m} \vec{T}_h \cdot cos(h\omega t + \delta_h) \right] \cdot e^{\frac{-t}{t_T}} \tag{11}$$

with $\omega = \frac{2\pi}{t_A}$, $t_A$ being the average time of one cycle within the one-minute interval $\Delta t$. $\delta_h$ is the phase, which within a simulation is chosen randomly being any number between zero and $2\pi$. $h$ specifies the number of harmonics contributing, with $m$ being the highest one. The maximal harmonic is identified from the Fourier transform of a subject's movement. $t_T$ denotes the time for the transient effect decreasing to $e^{-1}$. The transient effect averaged over the $n^{th}$ minute

is

$$\langle \vec{T}(n \cdot \Delta t) \rangle = \frac{1}{\Delta t} \cdot \int_{(n-1)\cdot\Delta t}^{n\cdot\Delta t} \left\{ \left[ \sum_{h=1}^{m} \vec{T}_h \cdot cos(h \cdot \omega \cdot t + \delta_h) \right] \cdot e^{-\frac{t}{t_T}} \right\} dt$$

$$= \left( \vec{T}_{\parallel} + \vec{T}_{\perp} \right) \cdot e^{-\frac{n\cdot\Delta t}{t_T}} \qquad (12)$$

Here and below $\parallel$ stands for the part of the vector pointing in the direction of the combined vectors of $\vec{T}(t)$ and $\vec{M}(t)$. $\perp$ indicates the vector parts perpendicular to the mutual direction.

6. When simulating the kinematics and comparing it with real life data, we need to include the measurement error–noise $\vec{N}(t)$—caused by the sensor characteristics. It can be obtained directly from measuring the output signals of the sensors at rest. The signal of an accelerometer is, subtracting the values caused by the earth's gravitational field, modeled as white noise.

$$\vec{N}(t) = RN[0, \sigma_{Sensor}] \cdot \begin{pmatrix} sin(\vartheta_s(t))sin(\varphi_s(t)) \\ sin(\vartheta_s(t))cos(\varphi_s(t)) \\ cos(\vartheta_s(t)) \end{pmatrix} \qquad (13)$$

Here $RN$ stands for a random normally distributed contribution with a mean value of $0 \, ^m/_{s^2}$ and a standard deviation $\sigma_{Sensor}$, which is the characteristic of the specific sensor. $\vartheta_s$ and $\varphi_s$ are randomly chosen to get a uniform distribution on the unique sphere. $\sigma_{Sensor}$ is calculated using Eq (13) and taking $\vec{N}(t)$ from the data recording of the sensors at rest.

## Model's characteristic constants

The main parameter for checking the model's validity is $\delta M$. It is the average distance between two data sets [23] and is calculated using Eq (1) by

$$\delta M = \frac{1}{\nu} \sqrt{\langle (\vec{K}(t) - \vec{K}(t_E))^2 \rangle}$$

$$= \frac{1}{\nu} \sqrt{\langle (\vec{A}(t) + \vec{M}(t) + \vec{F}(t) + \vec{T}(t) + \vec{C}(t) + \vec{N}(t)}$$

$$- \vec{A}(t_E) - \vec{M}(t_E) - \vec{F}(t_E) - \vec{T}(t_E) - \vec{C}(t_E) - \vec{N}(t_E))^2 \rangle$$

$$\approx \frac{1}{\nu} \sqrt{\langle (\vec{T}(t) + \vec{M}(t) - \vec{T}(t_E))^2 \rangle}$$

$$= \frac{1}{\nu} \left[ \langle T_{\parallel}(t) + M_{\parallel}(t) - T_{\parallel}(t_E) \rangle \right]$$

$$= \frac{1}{\nu} \cdot \langle T_{\parallel} \cdot \left[ e^{\frac{-t}{t_T}} - e^{\frac{-t_E}{t_T}} \right] + a_0 \cdot \left\{ \sqrt{\frac{(t_E - t)}{t_E}} + a_1 \cdot sin\left( a_2 \cdot 2\pi \frac{(t_E - t)}{t_E} \right) \right\} \rangle \qquad (14)$$

Here $\vec{A}(t) \equiv \vec{A}(t_E)$ by definition of an attractor as being identical at any cycle. The fluctuation together with the correction term do have almost identical averaged contributions close to zero at the different one-minute time intervals. The noise has contributions almost completely cancelling out within one minute because of its normal distributed character having a mean of zero. Therefore, the remaining input comes from the transient effect and the attractor morphing. We can calculate the length of the three lasting vectors. The remaining terms are the parallel contributions, all lying in the same direction at a given time, which can be written as a sum

of scalars. The subsequent equation allows us to write $\delta M$ depending on 5 constants $T_\|, t_T, a_0, a_1, a_2$, which are specified by curve fitting of the measurements. We use the software CurveExpert Professional 2.6.5, which uses the Levenberg-Marquardt algorithm providing the non-linear curve-fitting. While the three constants on the right describe the highly individual subject and task dependent morphing, the two constants on the left approximate the transient oscillation contributing to $\delta M$ at the beginning of a cyclic movement. $t_T$ depicts the time until the oscillation decreases to $1/_e$ of its original value $T$. The oscillation is negligible if $t_T \geq t_E$ (measuring time) since than the two exponential functions are almost equal to 1 resulting in these terms cancelling out. The values of the morphing and the transient effect do mix, which does not allow these two effects to be separated in all cases. Fortunately, there is a method to separate these two effects, which will be explained below. Altogether, we now have the nine constants $\phi\ t_E\ t_T\ t_A = \frac{2\pi}{\omega}\ T_\|\ b\ a_0\ a_1\ a_2$ determining our model. All definitions and the respective calculations/approximations are given to allow simulation of cyclic motion with the help of the attractors and the constants gained from the measured data. These simulations are naturally not identical to the original data, since the algorithm contains contributions of random processes.

## Data handling

Since further analysis required the collected 60-minute data block to be divided into 60-second intervals, a file splitter was applied to produce suitable single datasets. A raw data text-file contained thirteen columns: time and the acceleration as well as the gyro meter data in x, y and z direction for the left and the right foot, respectively. Afterwards an app called "Attractor", programmed with MATLAB was used to calculate the attractor data of every one-minute data set. Each attractor dataset contained 25 x n velocity/cadence-normalized data points: t, $x_{\text{left foot}}$, $y_{\text{left foot}}$, $z_{\text{left foot}}$, $x_{\text{right foot}}$, $y_{\text{right foot}}$, $z_{\text{right foot}}$, their standard deviations, standard errors, and gyroscope data. The functionality of the Attractor App is based on the attractor method developed by Vieten et al. [23] with the alteration of the attractor building process described above. The attractors were normalized for velocity in running and cadence (normalization factor v = cadence/10) for biking.

## Super attractor

A super attractor is by definition the average of all attractors of one subject, with the exclusion of any attractors that are to be compared to the super attractor. Also, no attractor influenced by the transient effect (usually those calculated from the data of the first 10 minutes of a measurement) is included. Specifically, for this study the super attractor was calculated for each participant from the collection of the final 50 minutes of each run independent of the data to be analyzed.

## Similarity analysis

For this procedure each attractor is recalculated having 500 data points by adjusting the sampling frequency using spline approximation. To find out how similar two movements are, we calculated the *recognition horizon* around each single attractor point, which is defined as the surface area at a distance equal to five standard deviations away from the attractor point. A test attractor is checked point to point if lying in- or outside the *recognition horizon* of the first attractor using another MATLAB procedure (Fig 1).

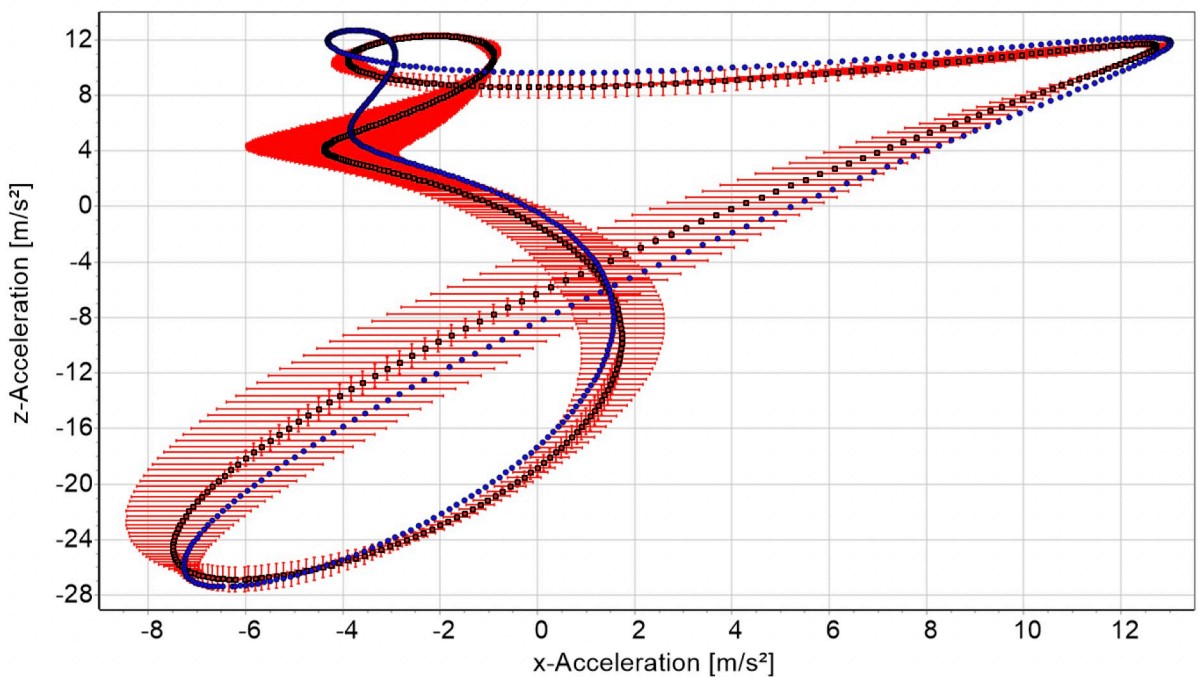

**Fig 1. Schematic two-dimensional depiction of the three-dimensional recognition horizon (red) and compared attractor (blue).**

Each measured or simulated minute over all running or cycling sessions (5 x 60 minutes) was checked against the respective *super attractor*. The *similarity rate* is defined the as percentage of data points lying within the *recognition horizons*.

## Separating the transient effect from morphing

To exclude the influence of the morphing as much as possible, we calculated a *super attractor* from 5 independent 1-hour-runs of each individual taken about 5 months before the actual measurements for running. For biking, as we did not have the data from months before, a *super attractor* was created out of four datasets to compare with the fifth one. Since our hypothesis was that an attractor is stable only within a given interval, the *super attractor* represents just one possible attractor configuration. It is important to note that these *super attractors* are independent of the 60 minutes data sets to be examined. Therefore, with the exception of the first minutes being influenced by the transient effect, the comparison should display results not varying much. And finally, the δM can be approximated by

$$\delta M_{without\ morphing} = c_0 + c_1 \cdot t + c_2 \cdot exp\left(\frac{-t}{t_T}\right) \tag{15}$$

As before, the constants are approximated applying the Levenberg-Marquardt algorithm through the software CurveExpert Professional 2.6.5. Here $c_0$ represents the strength of morphing. $c_1$ is the linear variation and is expected to be very small, since the distance between a super attractor and the attractors of a measurement should have very little variation with the exception of when the transient effect is active. Last $c_2$ denotes the strength of the transient effect.

## Simulation

For the simulation we created an app called "TrackSimulator" (accessible at http://www.uni-konstanz.de/FuF/SportWiss/vieten/CyclicMove/), available as Windows and macOS versions. It was created within MATLAB and is available as stand-alone solution without the need to install the MATLAB program. The app includes all the algorithms described above. To obtain the simulation the attractors of the tested subjects and their individual nine constants $\phi\ t_E\ t_T\ t_A\ T_\parallel\ b\ a_0\ a_1\ a_2$ serve as input for the app. We set the number of harmonics = 2 within the Eq (11), because those harmonics the majority of the signal's strength. Using the phase of the measurement within the simulation would give a good conformity between measurement and simulation. However, our first priority is about finding out about the variability of the cyclic motions. Therefore, the phase of the transient effect was chosen randomly.

## Subjects

A total of ten athletes, six female and four males, were tested in summer 2019. The running data (n = 5) were collected in Kreuzlingen, Switzerland (Nationale Elitesportschule Thurgau) whereas the cycling measurements (n = 5) took place at the University of Konstanz, Germany. All runners were active experienced recreational athletes. None had suffered any present injury, which could have impeded their performance. The cyclists were recruited from the local pool of university students. The only prerequisites were to be aged 18 years or older and able to run 60 minutes without reducing their initial pace or cycling at a moderate wattage over 60 minutes as regulated by their age, weight and training level [25], respectively. All participants were requested to fill out and sign an informed consent. The study was approved by the local Ethical Committee of the University of Konstanz, Germany under the RefNo: IRB19KN10-005.

## Equipment

To collect the necessary raw accelerometer data, two inertial sensors (RehaWatch by Hasomed, Magdeburg, Germany) were attached to both ankles by a hook-and-loop fastener during the runs; and on the proximal frontal part of the tibia (facies medialis) during the cycling tests. The sensors, MEMS—micro-electro-mechanical-system, have a size of 60x35x15 mm and weigh 35 g each. They function as a triaxial accelerometer, which we set up to a measurement interval of ±8 $g$, and a triaxial Gyroscope with up to 2000˚/$s$. The sampling rate was set to 500 Hz. Acceleration of the feet was measured in three dimensions (x, y, z) with data saved to a smartphone (Samsung Galaxy J5) using the app RehaGait Version 1.3.9 programmed by Hasomed (Magdeburg, Germany). All runs were performed on a treadmill (9500HR by Life Fitness, Unterschleißheim, Germany). The cycling measurements were undertaken on a cycle ergometer (ergoselect200, Ergoline, Bitz, Germany).

## Running data

The first session started with a short 5-minute warm up phase to get familiar with the treadmill and to determine an easy running pace associated with a BORG-scale of 3 [26] (Table 1).

**Table 1. Running speed of the subjects.**

|  | Subject 1 | Subject 2 | Subject 3 | Subject 4 | Subject 5 |
|---|---|---|---|---|---|
| **Running speed** | 10 km/h | 11 km/h | 10 km/h | 8.5 km/h | 8.7 km/h |

Table 2. Power output of subjects in biking.

|  | Subject 6 | Subject 7 | Subject 8 | Subject 9 | Subject 10 |
|---|---|---|---|---|---|
| Cadence | 90 rpm | 60 rpm | 65 rpm | 60 rpm | 60 rpm |
| Power output | 130 W | 60 W | 80 W | 50 W | 80 W |

The chosen running speed remained stable throughout all following test sessions each lasting 60 minutes. The participating athletes repeated the testing protocol in a time frame of approximately four weeks consisting of five testing days separated by at least 24 hours. The measurements were received from tri-axial accelerometers by a smartphone placed on a desk beside the treadmill to ensure undisturbed reception. Before the actual run, the participants set up the treadmill at 1% inclination (to simulate wind resistance) and their individual speed while waiting on the collateral standing area close to the treadmill belt. Once the chosen speed of the belt was reached the tester counted down from three to one before starting the data collection on the smartphone. At the same time, the runner jumped onto the treadmill belt and started immediately with running at the chosen pace over 60 minutes. This jumping movement, lasting approximately one second, was cut out of the data during the data management process, as it was a nonrunning-specific movement.

## Cycling data

Within four weeks, all cyclists repeated the testing protocol five times. Before the initial test day, the research group calculated the power and selected an appropriate seat position. All participants were tested at their preferred cadence (rpm = repetitions per minute), which the participants were able to hold within the interval of ±3 rpm over 60 minutes. Their power output conformed with an easy endurance workout and was defined using the athletes' age, weight and training level [25] (Table 2).

On each test day, the cyclists adjusted the seat and the handlebars as determined. The research assistant advised the athlete to hold the seating position and the cadence as stable as possible. The data collection was started by the tester immediately after the start signal caused the participant to pedal.

## Results

All input, measured data, and simulation results, had a sampling frequency of 500 Hz. Further procedures, including generating graphs, were done after filtering with a 'triple F low pass filter' [27] with a cutoff frequency of 10 Hz. $\phi$ $t_E$ $t_T$ $t_A = \frac{2\pi}{\omega}$ $T_{\parallel}$ $b$ $a_0$ $a_1$ $a_2$.

For the simulation, we used the constants $t_T$ $t_A = \frac{2\pi}{\omega}$ $T_{\parallel}$ $a_0$ $a_1$ $a_2$ taken from the measurements displayed below. The duration of simulation $t_E = 60$ $min$ was identical to the measurement's time. The random walk's strength was set $\phi = 100$ and the controlling constant at $b = 5$.

A graphic comparison between measurement and simulation gives a first impression of the model's power (Fig 2).

From $\delta M$ of the measurement we get the five constants $T_{\parallel}, t_T, a_0, a_1, a_2$. They are depending on the subject and on the specific movement. For our measurements we find the intervals of Table 3.

Similarity rates between measurements and simulation do show differences. This is expected since our model, in addition to containing deterministic parts, has random components as well. Important here is that the similarity analysis for running yields a gap between 50 and 56%, clearly separating same from different subject comparisons (Fig 3). All comparisons,

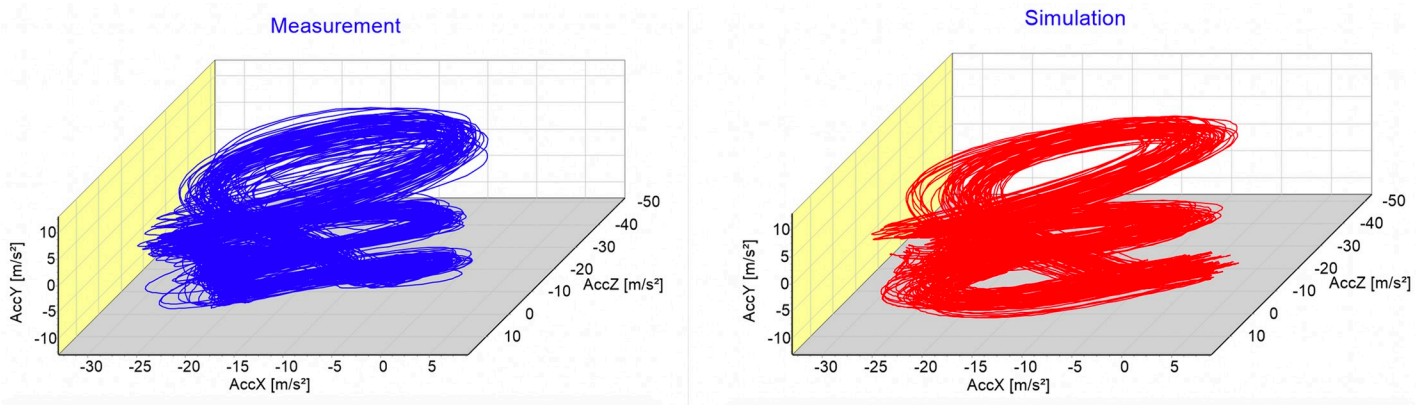

**Fig 2. Measured data (blue) and simulation results (red) of the first run of subject three.**

of measurements or simulations, between same subjects lie above the gap, comparisons between different subjects lie below.

For biking there is the same situation with a gap of 52 to 64 (Fig 4) clearly separating same from different subject comparisons. As before, all same subject comparisons lie above the gap, different subject comparisons below.

Values of $\delta M$ –Eq (14)–are influenced by morphing and the transient effect. A typical progression with both factors influencing $\delta M(t)$ is shown in Fig 5. In the first few minutes, the transient effect causes an increase/decrease, while morphing with its more moderate decline is visible afterwards. The differences between the measurement (blue) and the simulation (red) are caused by the transient effect and the "short time fluctuation". Here the starting conditions are largely random, causing differences at the beginning.

The difference between measurement and the simulation are caused by the transient effect and the "short time fluctuation". Here the starting conditions are largely random, causing differences at the beginning. The morphing of a specific measurement is imprinted into the simulation values via the Eq (3). A morphing effect is visible, if the analyzed minutes are from one uninterrupted measurement. The comparisons with the *super attractor* calculated from data independent of the actual numbers display random changes and the transient effect, but no morphing (Fig 6). Those data can be approximated using Eq (15), which allows calculation of the transient effect largely without the influence of morphing. $\delta M$ does not vary much with the only remarkable deviation at the beginning and up to about the $10^{th}$ minute.

Fig 6 shows $\delta M(t)$ for the five runs of subject 3, a representative where a substantial transient effect is prominently visible. Other subjects, especially the cyclists, show fewer or no exponential behavior at the beginning. Table 4 provides the time $t_T$ in minutes until the transient effect (TE) settles down to $e^{-1}$ of its initial value. This takes 4.3 minutes on average, where the cases without the transient effect are excluded.

The absolute height of $\delta M$ depends on the attractor's similarity compared with the independent *super attractor*. The following graphs, runs (Fig 7) and bike trials (Fig 8), show the mean

**Table 3. Overview of characteristic constants.**

| Constant | $T_\parallel$: Transient effect's strength | $t_T$: time for the transient effect decreasing to $T \cdot e^{-1}$ | $a_0$: morphing's strength | $a_1$: morphing's modulation strength | $a_2$: morphing's nonlinearity multiplier |
|---|---|---|---|---|---|
| Running | -3–10 | Individually given in | 1–8 | -0.4–0.5 | -0.3–1.8 |
| Biking | 0–10 | Table 4 | 0.5–13 | -0.3–0.3 | 0.3–4.2 |

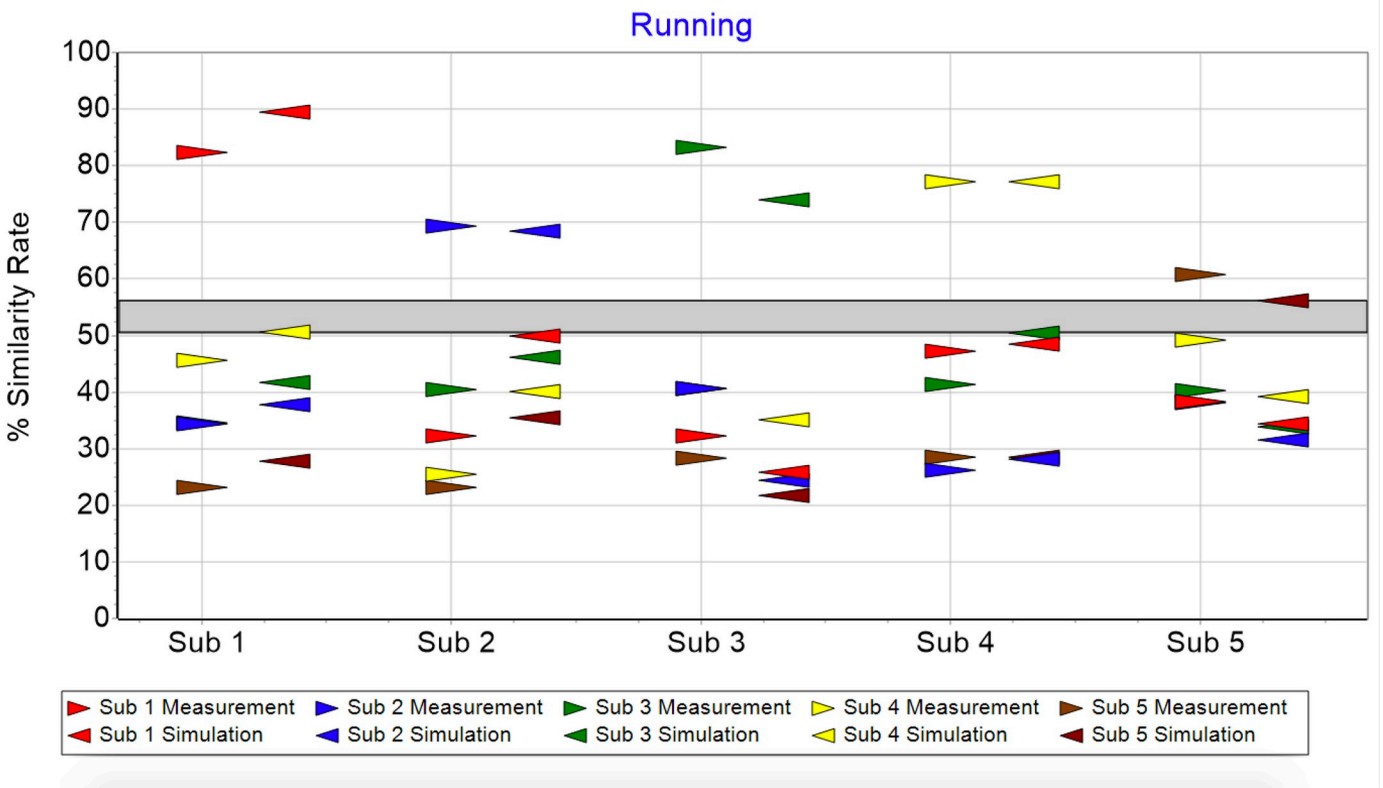

**Fig 3.** Similarity rate of running measurements (triangle pointing right) and simulations (triangle pointing left).

and the standard deviation of $\delta M$ for all subjects. The calculations are based on minutes 11 to 60, excluding the data influenced by the transient effect. Therefore, these values are a direct measure for morphing. For running $\delta M$ is in a range 2 to 5 m/s². Cycling displays values between 7 and 14 m/s² with one exception of a striking low $\delta M$ of about 1–1.4 m/s² for subject 10.

## Discussion

The purpose of this paper was to find a quantitative description of cyclic motion with the capacity to simulate individuals' characteristic movement. A model was proposed consisting of six contributing parts. Individual attractor, morphing, short time fluctuation, transient effect, control mechanism and sensor noise. Simulations based on this model showed the same distinctive variations as the measured data. In all cases the similarity analysis of same subjects produced higher results—$\delta_{run}^{same\ sub} > 55\%$ and $\delta_{bike}^{same\ sub} > 64\%$—compared with different subject combinations—$51\% > \delta_{run}^{different\ sub}$ and $52\% > \delta_{bike}^{different\ sub}$. Measurements of the respective simulations are clearly identifiable, confirming the model's suitability for describing cyclic motion. The nine constants together with the subject's attractor approximations are characteristic for a person's movement and the influence of the recording sensors.

As known from previous studies [21, 22] the influence of morphing and transient effect is small compared with the differences between individuals. While morphing is present in all trials, the transient effect is not observable in all cases (20 of 25 cases for running, 8 of 25 cases for biking). For biking, the transient effect is less prominent compared to running. We suspect the fixation of the legs with the foot connected to the pedal and the hip very much fixed onto

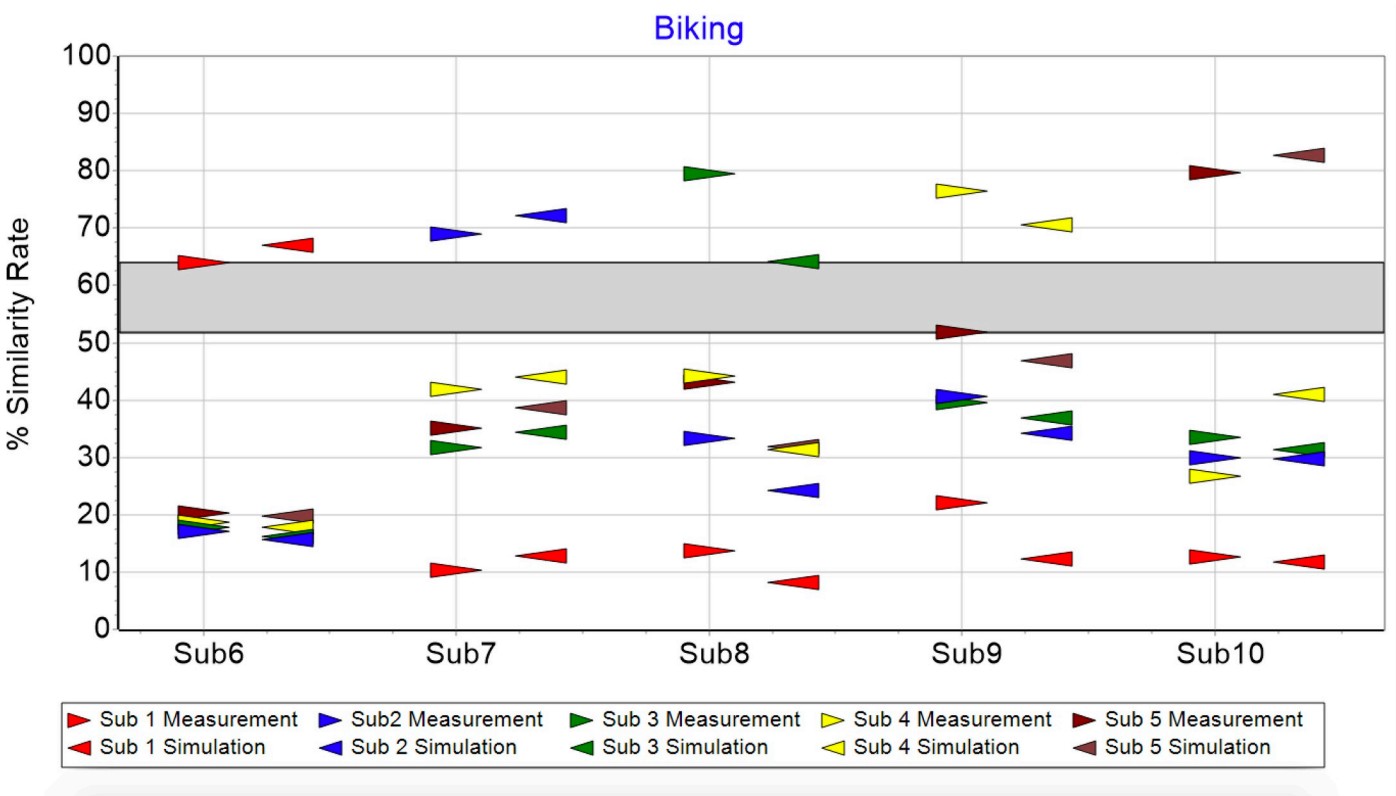

**Fig 4.** Similarity rate of biking measurements (triangle pointing right) and simulations (triangle pointing left).

the saddle, there is limited freedom in movement variation. The tibia position and its associated acceleration is often settled onto the attractor from the start onwards. A different situation is seen in running, where the kinematic chain is unfixed near the location of the accelerometer at the distal end of the tibia. Here the probability to start a movement close to the subject's attractor, resulting in no visible transient effect, is small. Interestingly, the most experienced runners show the least transient effect.

The comparison of a subject's attractors of a 1-hour measurement to an independent *super attractor* allows approximation of the magnitude of morphing. The maximal difference between attractors from independent measurements of one subject is restricted by the maximal possible morphing. Morphing can deform an attractor in many different ways, which most probably results in $\delta M$'s of comparable values. Therefore, results as shown in Figs 7 and 8 might represent good approximations of typical morphing magnitudes. Still, the determination of the attractor remains a challenging issue. In mathematical systems, like the famous "Lorenz map", the attractor is reached after the transient effect subsided. There, either a stable regular attractor is reached or a strange one is seen. Here, although data of the cyclic motion never completely reaches regularity, neither is the behavior completely chaotic. The regularity is, as mentioned before, good enough to discriminate between individuals.

Still the question remains, how to rate the attractors' differences, when attractor approximations are calculated by averaging the cycles of different time intervals. Does it simply mean that when doing the averaging over longer time periods these differences will almost completely vanish? Or, does it mean that attractors are changing with time, even if these changes are small? So far, we do not have enough data to answer this question with certainty.

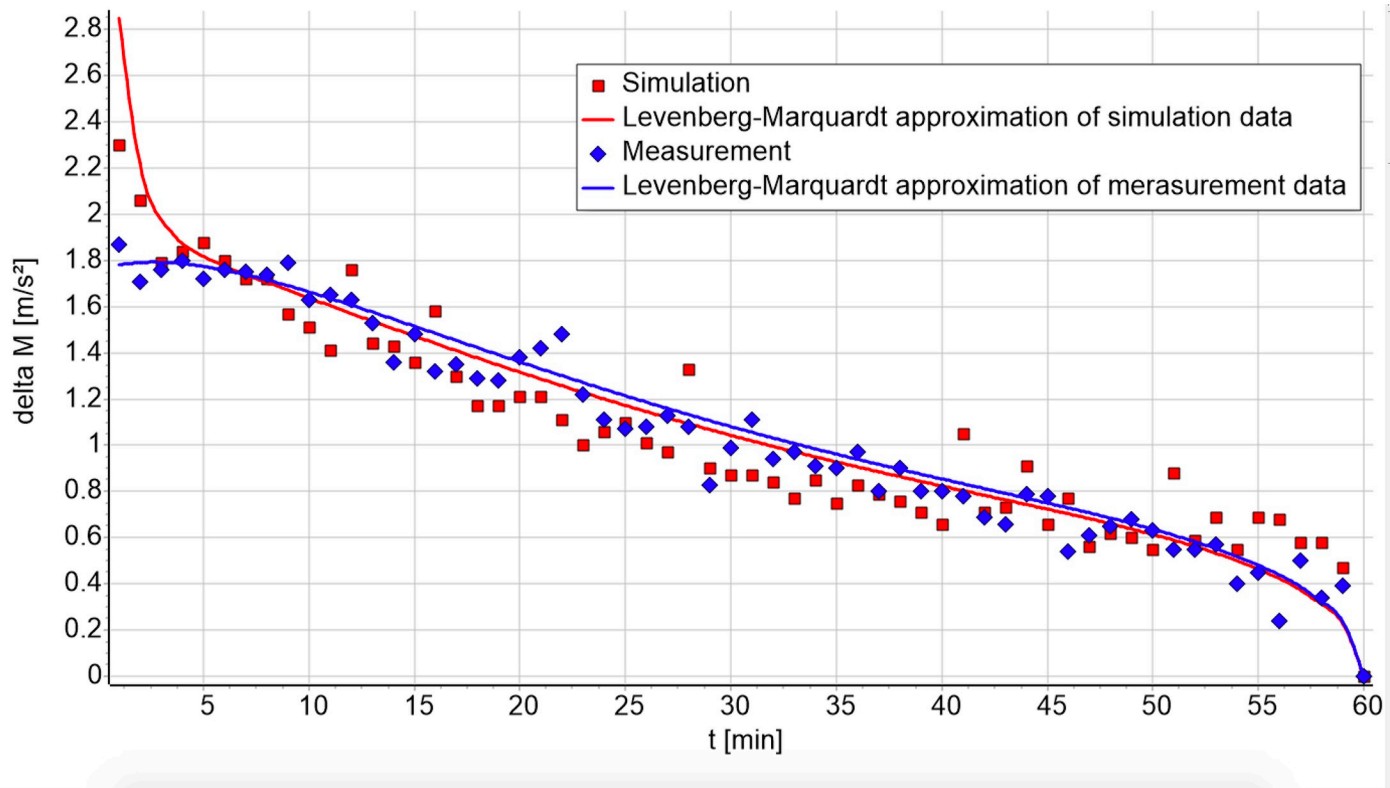

**Fig 5.** $\delta M(t)$ for the measurement (blue) of one run and the respective simulation (red).

However, from the results above we suggest that the second statement is more likely. There is a theoretical argument for this statement as well. While developing the mathematical description of cyclic motion, our first approach was without morphing. The idea was to have an attractor not dependent on time and the fluctuation based on a "random walk" characteristic only. This construct, however, did not allow describe the full data variability.

From a sport scientific view, the underlying components of the model are of particular interest in cyclic sports like running, cycling, swimming or rowing. Earlier work reports differences in subject-specific alternations in running patterns throughout prolonged activities like marathon running [28]. The latter authors state that competitive runners show a greater consistency of their subject-specific movement pattern compared to their recreational opponents, whose gait characteristics become significantly atypical halfway through the race. Further Clermont et al. [29] have demonstrated with their approach the ability to differentiate sex-and training level-specific subgroups based on acceleration data. An athlete with a extensive running experience combined with an increased mileage performs necessarily a higher number of strides leading to a more implanted and efficient movement pattern [30]. Thus, it can be assumed that the duration of the transient effect ends sooner combined with less deviations of the actual attractor contributed by the morphing effect. Should momentary accelerations still deviate from the morphed attractor, it can be expected that the control mechanism kicks in much sooner in athletes with a long-term training history. To check the mentioned expectations further application studies have to be conducted.

Altogether our model is capable describing cyclic motion quantitatively. Given the individual's attractor approximations and the subject specific constants, the output resulting from the

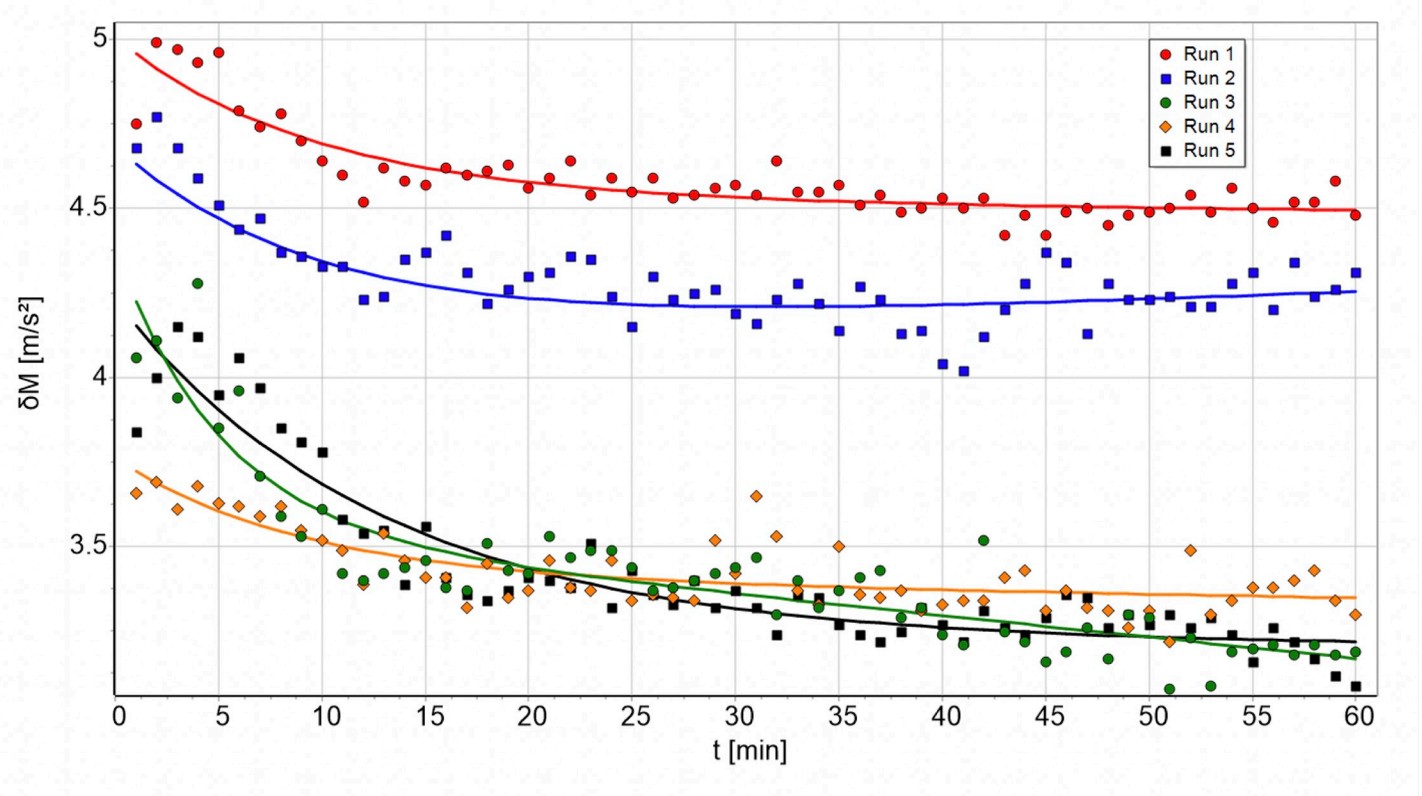

**Fig 6. Five runs of subject 3 compared to the subject's independent *super attractor*.** The lines represent the approximation as of Eq (15).

simulation is specific for the subject's particular movement. In addition, there are other aspects needing further attention. One is establishing a threshold for the similarity analysis to define the percentage when recognition is achieved. Many more measurements of a specific cyclic movement should allow determination of a suitable number by using the median method described by Vieten et al. [23]. Another limitation of the current approach is the focus on calculating δM, which depends on $\overrightarrow{T}_{\parallel}(t) + \overrightarrow{M}_{\parallel}(t)$, the parallel components only. Analyzing the full expression $\overrightarrow{T}(t) + \overrightarrow{M}(t)$ might allow further insight.

## Conclusion

This paper is a "proof of concept" showing cyclic motion can be described with the mathematical model introduced. Moreover, the simulation based on the developed model is capable of generating numbers displaying the same structure and behavior as the measurement.

**Table 4. The time $t_T$ [min] by which the transient effect (TE) reduces to $e^{-1}$ of its start value.**

| Sub | Run 1 | Run 2 | Run 3 | Run 4 | Run 5 | Sub | Bike 1 | Bike 2 | Bike 3 | Bike 4 | Bike 5 |
|---|---|---|---|---|---|---|---|---|---|---|---|
| **Sub 1** | 1.0 | 5.1 | 3.9 | No TE | No TE | **Sub 6** | No TE | No TE | No TE | No TE | No TE |
| **Sub 2** | 5.0 | No TE | No TE | 5.0 | No TE | **Sub 7** | No TE | No TE | No TE | No TE | No TE |
| **Sub 3** | 9.7 | 10.0 | 5.1 | 9.0 | 12.9 | **Sub 8** | 3.4 | 4.3 | 3.4 | 2.8 | 3.5 |
| **Sub 4** | 1.7 | 1.8 | 0.9 | 5.5 | 1.0 | **Sub 9** | No TE | No TE | No TE | No TE | No TE |
| **Sub 5** | 1.9 | 2.8 | 3.7 | 2.3 | 3.5 | **Sub 10** | No TE | 8.2 | 1.5 | 1.4 | No TE |

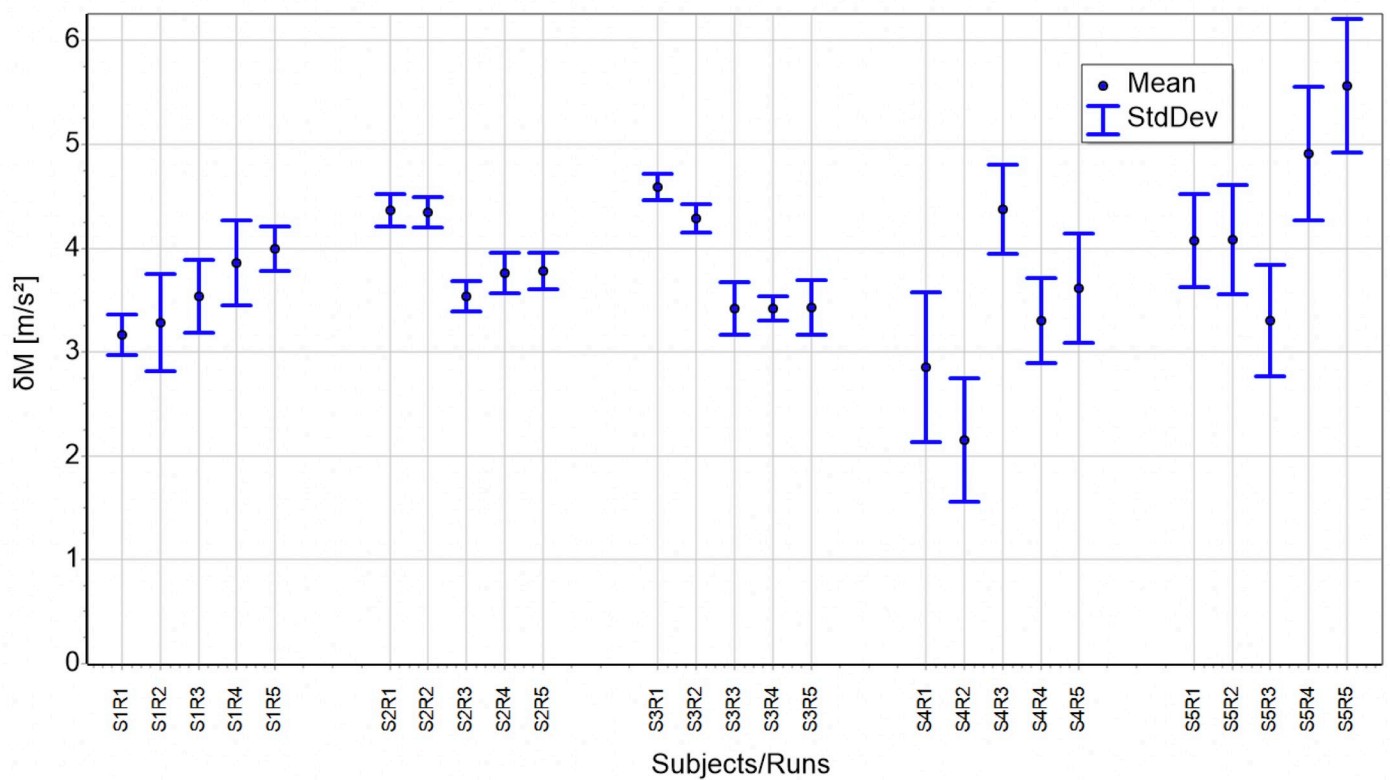

**Fig 7. All 5 runs of all 5 subjects compared to their personal, but independent super attractor for minutes 11 to 60.**

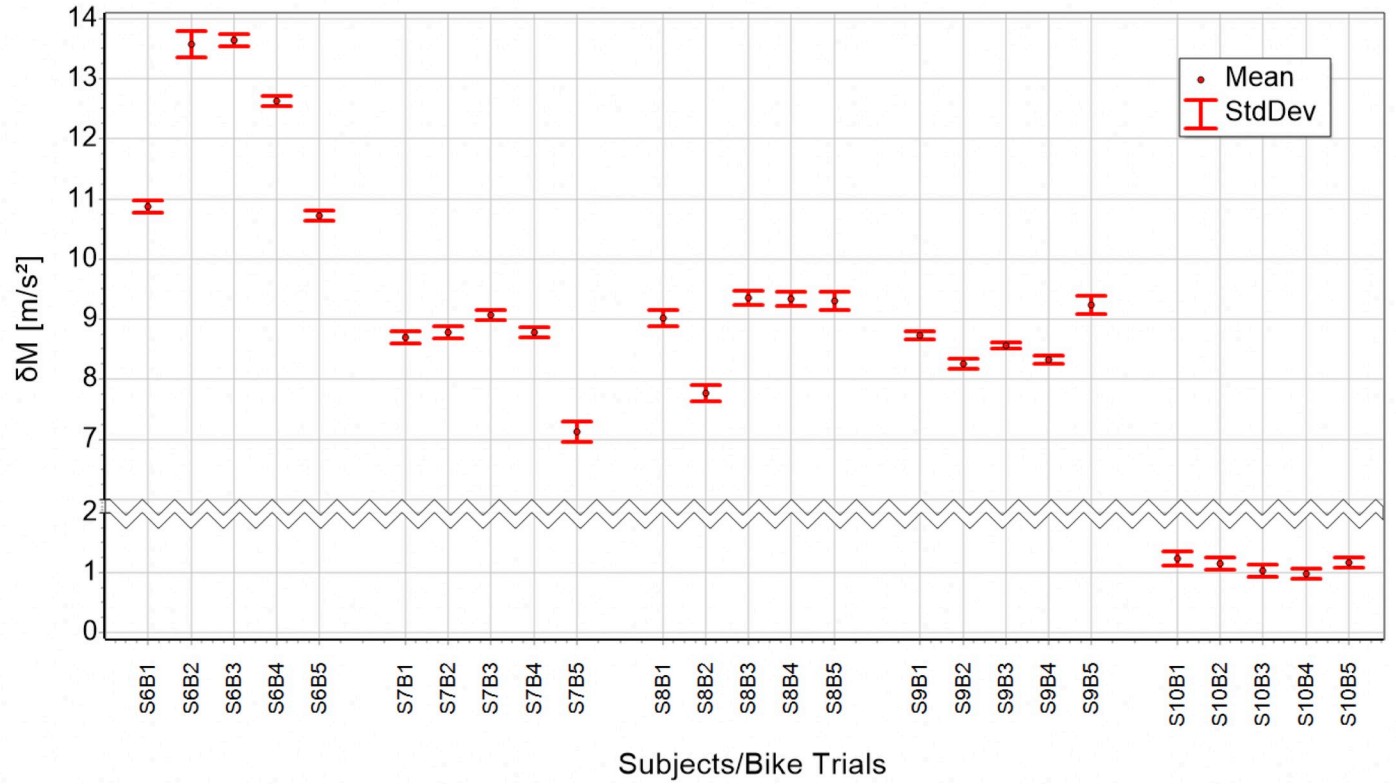

**Fig 8. All 5 bike trials of all 5 subjects compared to their personal, but independent super attractor for minutes 11 to 60.**

Applications are conceivable in the areas medical diagnostics, performance assessment, subject recognition, and robotics. For diagnostics, our group has previously developed and used a fatigability scale for multiples sclerosis patients [31, 32]. The new model however, allows description of the transition between normal and fatigability conditions more precisely by considering morphing. In terms of performance assessment, the results of Figs 7 and 8 suggest morphing's magnitude is different depending on the specific subject. This might be correlated to athletes' performance levels, using stable running patterns throughout prolonged physical activities. Further, it might allow deeper insight into the dependencies of parameters such as gender, training history and anthropometric attributes. Figs 3 and 4 –the arrows above the gap —show with the help of the similarity rate, that it is possible to find measurement/simulation combinations belonging to the same subject. This fact and some preliminary analyses suggest subject recognition is possible though attractor comparison. Here the attractor of a measurement is compared with a database of attractors. Finally, bipedal robots' movement might profit from our model by comparing the specific values of the characteristic constant, as well as the specific form of the attractors between humans and robots.

## Acknowledgments

We thank all subjects who participated in the study.

## Author Contributions

**Conceptualization:** Manfred M. Vieten, Christian Weich.

**Data curation:** Manfred M. Vieten, Christian Weich.

**Formal analysis:** Manfred M. Vieten, Christian Weich.

**Funding acquisition:** Manfred M. Vieten.

**Methodology:** Manfred M. Vieten, Christian Weich.

**Project administration:** Manfred M. Vieten.

**Software:** Manfred M. Vieten, Christian Weich.

**Supervision:** Manfred M. Vieten.

**Validation:** Manfred M. Vieten, Christian Weich.

**Visualization:** Manfred M. Vieten.

**Writing – original draft:** Manfred M. Vieten.

**Writing – review & editing:** Manfred M. Vieten, Christian Weich.

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
