## [Decision Letter · Decision Letter 0]

12 Dec 2019

PONE-D-19-29994

The kinematics of cyclic human movement

PLOS ONE

Dear Dr. Vieten,

Thank you for submitting your manuscript to PLOS ONE. After careful consideration, we feel that it has merit but does not fully meet PLOS ONE’s publication criteria as it currently stands. Therefore, we invite you to submit a revised version of the manuscript that addresses the points raised during the review process.

We would appreciate receiving your revised manuscript by Jan 26 2020 11:59PM. To enhance the reproducibility of your results, we recommend that if applicable you deposit your laboratory protocols in protocols.io, where a protocol can be assigned its own identifier (DOI) such that it can be cited independently in the future. For instructions see: http://journals.plos.org/plosone/s/submission-guidelines#loc-laboratory-protocols

We look forward to receiving your revised manuscript.

Kind regards,

Nizam Uddin Ahamed, PhD

Academic Editor

PLOS ONE

2. Your ethics statement must appear in the Methods section of your manuscript. If your ethics statement is written in any section besides the Methods, please move it to the Methods section and delete it from any other section. Please also ensure that your ethics statement is included in your manuscript, as the ethics section of your online submission will not be published alongside your manuscript.

Reviewers' comments:

Reviewer's Responses to Questions

**Comments to the Author**

1. Is the manuscript technically sound, and do the data support the conclusions?

Reviewer #1: Partly

Reviewer #2: Partly

Reviewer #3: Yes

2. Has the statistical analysis been performed appropriately and rigorously? 

Reviewer #1: No

Reviewer #2: I Don't Know

Reviewer #3: N/A

3. Have the authors made all data underlying the findings in their manuscript fully available?

Reviewer #1: Yes

Reviewer #2: Yes

Reviewer #3: Yes

4. Is the manuscript presented in an intelligible fashion and written in standard English?

Reviewer #1: No

Reviewer #2: Yes

Reviewer #3: Yes

5. Review Comments to the Author

Reviewer #1: The present study provides a mathematical model for simulating the kinematics of human cyclic movement based on a main spline equation called attractor and subsequent modifying polynomials/noise functions with their factors being identified experimentally. The idea sounds interesting and practical; however, the presentation of manuscript and results have significant limitations that need to be addressed before being considered for publication. The main issues are:

1) The authors have provided intensive mathematical models without a proper prove and rational for their selection. In addition, the equations are poorly represented with many undefined elements across the paper. This highly affects the readability of this manuscript and is necessary to be revised thoroughly.

2) Abstract and Results: no overall quantified accuracy/performance result (e.g. averaged across all subjects) is reported to compare the simulations and measurements.

3) Simplification of Equation 14 is vague and no mathematical proof is provided.

4) The simulation app (TrackSimulator) seems to have problem in running at least on my computer (macOs).

5) Super Attractor section: what does (minute x attractor time/acceleration/deviation/standard error/gyroscope data x average number of attractor points) show? Is it a vector? It is confusing.

6) Eq. 2: the authors have claimed that they used a fixed number of data points instead of a fixed time interval. This shows that the sampling frequency was not consistent otherwise this should not matter. Please explain.

7) Eq. 12: how do you define the direction of these vector systems? The functions are in terms of “t” which is a scalar and no geometrical direction is introduced. This is a bit confusing.

8) Low quality figures. Fig. 2: what quantity does it show? Axis names are not visible.

9) Fig. 6: how is the super attractor function identified? From the same data that are shown in the figure? How do you reproduce the function and prove its functionality in different walking conditions, e.g. walking up/down the stairs?

10) Table 3: why are there many empty cells in this table? No transient effect present? Please explain.

11) The organization of paper is poorly handled as there are only one paragraph for introduction and discussion! Furthermore, I found several grammatical issues throughout the manuscript including:

a. Model-number 1-line 9: circle -> cycle?

b. Model-number 6-line 2: real live -> real life?

c. Introduction-line 35: gives inside -> gives insight?

12) The participants were regarded as athletes in the paper, but the general requirement of recruiting subjects does not reflect them being an athlete.

Reviewer #2: The purpose of the paper is to expound a model of cyclic human behavior. It is evaluated for 10 individuals, 5 for running and 5 for cycling. The major part of the work is the evaluation of the simulated model. The readability of the results leaves a lot to be desired as from some figures (3,4) it is impossible to see a lot.

A clear conclusion is missing. While there is a discussion present, a clear conclusion is not given and therefore the reader is left alone to ponder a lot of questions set up in the discussion, and to come to the conclusion by oneself. This is what is most missing – a conclusion saying that the model is better than some other models and that it allows for something that was previously not allowed – all stated explicitly. Also, this implies some comparison, which is not present.

Dynamical systems are not at all discussed in the paper, or just barely. Also, working with the phase of the motion would help a lot. Also, some deep neural networks approach that can be used to classify people based on data, and therefore somehow probably also predict the data are not mentioned, but surely they are out there.

Given the amount of work (measurements) put into this paper, it is a bit too bad that it lacks in the presentation and comparison, but it is what still gives a chance to the paper.

Level of English could be improved (is not bad, but at some sentences I was really scratching my head wondering what was meant)

Some specific comments

Abstract starts with some claims on “roughly be divided into the categories theory or data driven”, which are totally unfounded – ok, citing is bad in the abstract, but then write it differently. Such claims – without data or at least citations, are totally useless and just say: I said this, now believe it! Not good practice

and how such a developed model can be tested? -> I really do not understand this sentence and what you are trying to ask yourself

No wonder that -> this does not seem as an appropriate form for a scientific paper…

this model does allow -> ALLOWS

There exist two types of models –> two types of models of what?

What are the colors of the arrows in Fig 3 and 4? And they don’t really match for some subjects…

Reviewer #3: In this work, the authors developed a mathematical model to describe cyclic human movement. They then collected data from both running and biking, and fit the model based on subject-specific measurements. Simulation of the fitted models generates movements that are quite similar to measurement, suggesting that the model is able to, at least generally, describe the motion. Overall, I have no major reservations about the validity of this work but have a number of recommendations that would improve the overall readability of this paper. Please see my attachment.

6. PLOS authors have the option to publish the peer review history of their article (what does this mean?). If published, this will include your full peer review and any attached files.

Reviewer #1: Yes: Amir Baghdadi

Reviewer #2: No

Reviewer #3: No

---

## [Author Response · Author response to Decision Letter 0]

24 Jan 2020

Responses to all remarks in the decision letter are included in the “Response to Reviewers.docx”

---

## [Decision Letter · Decision Letter 1]

19 Feb 2020

The kinematics of cyclic human movement

PONE-D-19-29994R1

Dear Dr. Vieten,

We are pleased to inform you that your manuscript has been judged scientifically suitable for publication and will be formally accepted for publication once it complies with all outstanding technical requirements.

With kind regards,

Nizam Uddin Ahamed, PhD

Academic Editor

PLOS ONE

Additional Editor Comments (optional):

Reviewers' comments:

Reviewer's Responses to Questions

**Comments to the Author**

1. If the authors have adequately addressed your comments raised in a previous round of review and you feel that this manuscript is now acceptable for publication, you may indicate that here to bypass the “Comments to the Author” section, enter your conflict of interest statement in the “Confidential to Editor” section, and submit your "Accept" recommendation.

Reviewer #1: All comments have been addressed

Reviewer #3: All comments have been addressed

2. Is the manuscript technically sound, and do the data support the conclusions?

Reviewer #1: Yes

Reviewer #3: (No Response)

3. Has the statistical analysis been performed appropriately and rigorously? 

Reviewer #1: Yes

Reviewer #3: (No Response)

4. Have the authors made all data underlying the findings in their manuscript fully available?

Reviewer #1: Yes

Reviewer #3: (No Response)

5. Is the manuscript presented in an intelligible fashion and written in standard English?

Reviewer #1: Yes

Reviewer #3: (No Response)

6. Review Comments to the Author

Reviewer #1: (No Response)

Reviewer #3: (No Response)

7. PLOS authors have the option to publish the peer review history of their article (what does this mean?). If published, this will include your full peer review and any attached files.

Reviewer #1: No

Reviewer #3: No

---

## [Editor Report · Acceptance letter]

21 Feb 2020

PONE-D-19-29994R1 

The kinematics of cyclic human movement 

Dear Dr. Vieten:

I am pleased to inform you that your manuscript has been deemed suitable for publication in PLOS ONE. Congratulations! Your manuscript is now with our production department. 

With kind regards,

on behalf of

Dr. Nizam Uddin Ahamed 

Academic Editor

PLOS ONE